# Deciphering the Complex Molecular Pathogenesis of Myotonic Dystrophy Type 1 through Omics Studies

**DOI:** 10.3390/ijms23031441

**Published:** 2022-01-27

**Authors:** Jorge Espinosa-Espinosa, Anchel González-Barriga, Arturo López-Castel, Rubén Artero

**Affiliations:** 1University Research Institute for Biotechnology and Biomedicine (BIOTECMED), Universidad de Valencia, 46100 Valencia, Spain; jorge.espinosa@uv.es (J.E.-E.); ruben.artero@uv.es (R.A.); 2Translational Genomics Group, Incliva Biomedical Research Institute, 46010 Valencia, Spain; 3Centre de Recherche en Myologie, Inserm, Institut de Myologie, Sorbonne Université, 75013 Paris, France; anchel.gonzalez-barriga@inserm.fr

**Keywords:** transcriptomics, proteomics, alternative splicing, alternative polyadenylation, expanded CUG repeats, gene expression, therapies, myotonic dystrophy, RNA-binding proteins, RNA metabolism

## Abstract

Omics studies are crucial to improve our understanding of myotonic dystrophy type 1 (DM1), the most common muscular dystrophy in adults. Employing tissue samples and cell lines derived from patients and animal models, omics approaches have revealed the myriad alterations in gene and microRNA expression, alternative splicing, 3′ polyadenylation, CpG methylation, and proteins levels, among others, that contribute to this complex multisystem disease. In addition, omics characterization of drug candidate treatment experiments provides crucial insight into the degree of therapeutic rescue and off-target effects that can be achieved. Finally, several innovative technologies such as single-cell sequencing and artificial intelligence will have a significant impact on future DM1 research.

## 1. Introduction

Myotonic dystrophy type 1 (DM1), an inherited neuromuscular disease, is the most common type of muscular dystrophy in adults, with a reported prevalence of up to one in every 2100 births [1,2]. DM1 is a multisystem disorder with diverse symptoms, including muscle hyperexcitability (myotonia), progressive muscle wasting, cardiac arrhythmias, insulin resistance, gastrointestinal dysfunctions, posterior iridescent cataracts, and neuropsychiatric disturbances [3]. The disease is caused by an expanded trinucleotide (CTG) repeat in the 3-prime untranslated region (3′-UTR) of the DM1 protein kinase (*DMPK*) gene. Disease severity varies with the number of CTG repeats: unaffected individuals carry 5 to 37 triplets, mildly affected persons between 50 and 150, patients with classic DM1 from 100 to 1000, and those with onset at birth can have more than 2000 repeats [4]. The primary molecular mechanism causing this disorder is the toxicity generated by expanded CUG repeats present in the 3′-UTR of mutant *DMPK* transcripts, which alters the function of various downstream effectors, triggering gene deregulations through alterations in transcription [5], translation [6], gene silencing [7], alternative splicing [8], and polyadenylation [9]. In DM1, the most studied families of RNA-binding proteins (RBPs) affected by mutant *DMPK* transcript toxicity are the CUGBP-ETR-3-like (CELF) and Muscleblind-like (MBNL) factors, both controlling the inclusion of alternative exons in several transcripts according to tissue and developmental state cues [10]. Currently, there is no cure available to halt or slowdown DM1 progression. Therefore, therapeutic advances are urgently needed since patient medical care is limited to clinical management and symptomatic treatment, such as anti-diabetic or anti-myotonic drugs, rehabilitation therapy, or surgery, which, despite improving the quality of life of the patients, are not definitive treatments against the root causes of the disease [11]. New and promising treatments are being developed to suppress or even eliminate the molecular effects of DM1, such as toxic RNA degradation or MBNL expression enhancers [12].

The MBNL family comprises the MBNL1, MBNL2, and MBNL3 RNA metabolism regulators, whose expression levels are tightly regulated by the developmental stage in each tissue. *MBNL1* and 2 are broadly expressed, but *MBNL1* is the paralog that serves a primary role in most tissues, except for the brain, where *MBNL2* is predominantly detected, whereas *MBNL3* expression is more restricted and has been related to muscle-cell differentiation inhibition, aging, and regeneration [13,14,15,16]. The CELF RBP family comprises six members, divided into two groups according to their expression level. CELF1 and 2 are highly expressed in various tissues, such as skeletal muscle, brain, and heart, constituting the first and most studied group. CELF3–6 are mostly present in neurons, and CELF6 can also be found in kidneys and testes [17,18].

Alternative splicing (AS) is an RNA-processing mechanism that removes sequences (introns) between splicing sites in pre-mRNAs to process mRNAs [19]. It is mediated by the spliceosome and heavily regulated by different sequence motifs, which are recognized by RBPs that can act as repressors or enhancers of each splicing site. This process contributes crucially to proteome diversity by allowing a single gene to code more than one mRNA and potentially have more than one function [20]. Polyadenylation is the last key step in RNA maturation, which involves the cleavage of nascent pre-mRNA 3′-end and addition of a poly(A) tail at the cleavage site, which plays a leading role in the translation efficiency, stability, and localization of mRNAs. If a gene possesses more than one poly(A) site, the difference in usage of those sites is called alternative polyadenylation (APA) [21,22]. In DM1, both AS and APA processes have been reported to be altered and to play an essential role in pathogenesis [9].

Various disease models have been developed to investigate the pathogenic mechanisms associated with DM1. Some of the most commonly used in omics studies are: (1) the HSA^LR^ mouse model, a transgenic mouse carrying a genomic fragment of the human skeletal actin (*ACTA1*) gene with ~250 CTG repeats in the 3′-UTR, which is expressed in the skeletal muscle only. This model recapitulates RNA toxicity alterations by displaying myopathy, centrally located nuclei in muscle fibers, and AS defects [23]. (2) The DMSXL mouse model is a transgenic line carrying 45 kb of the human genome from the DM1 locus with >1000 CTG repeats. DMSXL mice reproduce key molecular hallmarks of DM1, such as nuclear foci in most tissues (except epithelia), RNA mis-splicing in multiple tissues, and relevant muscle, cardiac, neuron, and glial cell phenotypes [24,25,26]. Finally, (3) patient-derived immortalized myoblasts (PDIM) with >2000 CTG repeats can be used for in vitro experiments. They show ribonuclear aggregates that colocalize with MBNL1 and splicing defects in key genes such as *ATP2A1*, *BIN1*, *INSR*, *LDB3*, *MBNL1*, and *TNNT2* [27]. Other less frequently used or recently developed models will be discussed in the corresponding section.

In recent years, new advances in and the widespread application of omics technologies have enabled cost-efficient, high-throughput, in-depth analyses of transcripts, proteins, and other molecules, creating an unprecedented body of knowledge [28] and enabling advances in personalized medicine [29], cancer research [30], and rare diseases [31,32]. Herein, we reviewed the literature on DM1, since the number of omics studies far exceeds those involving Myotonic Dystrophy Type 2 (DM2), focusing particularly on the tools used to elucidate the pathological mechanisms involved, to broaden the scope of our understanding of this complex multisystem disorder.

## 2. Omics Studies in DM1

Transcriptomics is the study of the full range of transcripts at the RNA level of a biological system [33]. Tools such as microarrays, RNA-Sequencing (RNA-Seq), high-throughput sequencing of RNA isolated by crosslinking immunoprecipitation (HITS-Clip), and their corresponding variations, have been extensively used to study transcriptome complexity in DM1. Splice-sensitive microarrays have been employed to identify disease-related splicing alterations and describe specific deregulations of isoform expression, while hundreds of target sequences have been established for both MBNL and CELF proteins using HITS-Clip. More recently, RNA-Seq has been used to determine gene expression and splicing changes in various DM1 tissues and disease models. All these studies will be covered in more detail in the following sections. Transcriptomics, however, does not provide reliable information about the fate of a given gene product after translation. Therefore, analysis of the proteome is essential to elucidate the extent of DM1 post-transcriptional dysregulation in each cell type, tissue, developmental stage, or disease condition. In this regard, mass spectrometry analysis represents the most potent proteomics-based technology to quantify steady-state protein levels (influenced by features such as translation rates or increased degradation) and identify new candidates mediating pathological DM1 mechanisms [34].

The broad DM1 symptomatology impinges particularly on three critical body systems: skeletal and cardiac muscles and the brain. Together, these tissues comprise the focus of most omics studies done in DM1 so far, addressed specifically below (Figure 1).

### 2.1. Omics Studies on Skeletal Muscle

One of the most supported hypotheses in DM1 is that MBNL proteins are responsible for a large fraction of the aberrant AS patterns observed in muscle. Initial work with splicing sensitive microarray analysis found that 80% of approximately 200 alternative isoform changes observed in HSA^LR^ mice quadriceps were reproduced in the same muscle of mice lacking functional MBNL1 proteins (MBNL1^ΔE3/ΔE3^ [35]), leaving 20% of isoforms affected in an MBNL1-independent but CUG-expansion-dependent manner [36,37]. A later study performed in the same model, using RNA-Seq, identified almost four times the number of misregulated AS events than were found by microarrays [38]. A complementary RNA-Seq analysis with C2C12 immortalized mouse myoblast cells, silenced for *MBNL1*, *MBNL2*, or both, observed that mis-splicing severity was greater in doubly silenced cells than in each individual condition. These results indicated some redundancy in MBNL1 and 2 functions. Furthermore, a HITS-Clip experiment with several mouse tissues (brain, heart, and *vastus lateralis*) and C2C12 cells helped map the *MBNL* binding sites precisely. This study revealed that skeletal muscle had the most MBNL1 CLIP sites, and that these sites showed a defined pattern around alternative exons across all samples and tissues, suggesting that splicing activation or repression depends on where MBNL1 binds. Specifically, two repression peaks were found, one located right on the cassette exon and another upstream of the 3′-splice site, consistent with the blocking of exon enhancer elements or intronic elements (required for effective intron removal as branch site recognition by the U2AF65 protein), respectively [39]. However, additional intronic peaks were found in other regions, suggesting that repression may occur by additional mechanisms. An activation-associated CLIP peak was found 120 bases downstream of the 3′ end of the alternative exon. Thus, while MBNL1 binding to the alternative exon or nearby upstream sequences represses exon inclusion, binding to downstream sites enhances it [38].

A different RNA-Seq study analyzing muscle samples of a compound loss of MBNL1^ΔE3/ΔE3^; MBNL2^+/−^ mouse (so-called, MBNL^3/4^KO) showed a high correlation of splicing alterations between MBNL^3/4^KO, MBNL1^ΔE3/ΔE3^, and HSA^LR^, indicating a major role for MBNL sequestration in murine models of this disorder [40]. MBNL binding motif analysis was also performed, and the findings fit with the previously described patterns [38] in which binding motifs in downstream introns cause increased exon skipping, while those in upstream introns cause increased exon inclusion in DM1. Furthermore, 35 AS events, highly representative of DM1, were analyzed using a novel methodology called “targeted splice sequencing”, which allows for more efficient and accurate analysis of known splicing alterations. In this study, characteristic AS events were analyzed at different developmental stages on wild type, HSA^LR^ and MBNL^3/4^KO mice. Clear developmental transitions for most events across embryonic and adult samples were observed, suggesting that mis-splicing in DM1 reflects a reversion to fetal or neonatal AS patterns in muscle, as has been extensively reported in previous studies [8,38,40]. In another study, to further investigate alterations in developmental stages, human embryonic stem cells (hESC) were subjected to myogenic differentiation and analyzed by RNA-Seq to find evidence of differences in two developmental transitions using control and DM1 samples from hESC, myoblasts, and myotubes [41]. No core myogenic regulatory genes were altered in the hESC to myoblast transition, while a considerable number of the gene expression changes present in control myoblast to myotubes transitions were not observed in DM1 samples. Gene set enrichment analysis of these genes found an interferon-alpha response belonging to interferon type I and altered mTORC1 signaling pathways.

Despite major advances from in vitro and in vivo models, the DM1 field lacked a transcriptome dataset that would serve as a benchmark to compare to subsequent experimental analyses. To this end, www.dmseq.org (accessed date 20 December 2021) was created as part of a major RNA-Seq work that sequenced various human samples from biopsies and autopsies, including *tibialis anterior* samples of 44 DM1 patients and 11 unaffected controls [42]. While most of these samples were sequenced to a depth of at least 41 million reads, a subset of 50 were sequenced extremely deeply (>200 million reads). Combining all the data, which required specific approaches for the analysis because of the different depths, provided an accurate assessment of gene expression and AS for a significant portion of the transcriptome. Notably, RNA-level data could be correlated with patients’ ankle dorsiflexion strength measurements, finding a high correlation between strength and MBNL activity (defined as free MBNL protein), which was estimated by the degree of detected MBNL-dependent AS alteration (validated in a previous study using human cells) [43]. Using this value, a significant correlation was found between MBNL activity and AS of key DM1 genes such as *CLCN1* and *CLASP1* and various alternative last exon inclusions. Furthermore, AS alterations found in biopsies correlated significantly with known DM1 splicing biomarkers [44]. Notably, AS alterations in the quadriceps were lower than in *tibialis* biopsies. In the case of autopsies, a lower degree of altered AS pattern was also observed in the quadriceps when compared with other distal muscles, such as the *deltoids*, *gastrocnemius*, or *soleus*, providing a molecular explanation for the distal to proximal pattern of muscle weakness and wasting found in DM1 patients (milder phenotype shown in proximal muscles) [45]. The AS findings of this study replicated previous observations regarding DM1 muscle RNA metabolism, such as the abnormal inclusion of exon 7a or retention of intron 2 in muscle-specific chloride voltage-gated channel 1 (*CLCN1*) transcripts, both triggering nonsense-mediated decay resulting in a severe reduction in functional CLCN1 [46].

To evaluate the full extent of AS deregulation in DM1, another study used whole human exome microarrays to compare skeletal muscle samples of DM1 with those of other neuromuscular disorders: dystrophy type 2 (DM2), Duchenne muscular dystrophy, Becker muscular dystrophy, and Tibial muscular disease [47]. Compared to unaffected controls, 362 AS alterations were common among all diseases, and a clear tendency to gene upregulation was observed in both DM1 and DM2 compared to the other diseases. Thus, while all these disorders arise from different mutations, they share common downstream pathomechanisms in pathways related to RNA metabolism (particularly AS), growth arrest, and DNA damage-inducible genes [47].

Hundreds of genes presenting APA have been reported in DM1 muscle as part of previous RNA-Seq work [38]. A later study evaluated APA transitions in mouse embryonic fibroblasts (MEF) MBNL1^ΔE3/ΔE3^ and MBNL2^ΔE2/ΔE2^ double KO (DKO) and in MBNL null MEF (DKO/3KD) conditions, the last obtained by siRNA-induced MBNL3 silencing in the DKO background [48]. Microarrays and RNA-Seq from a library enriched in transcripts with poly(A) tails (polyA-Seq) were used to compare control vs. DKO and control vs. DKO/3KD. Over 4000 APA events were found altered in the first comparison, while around a thousand more were found in the second, indicating that MBNL proteins are essential for normal MEF APA regulation. To further understand the role of MBNL3, control and DKO MEFs were analyzed with HITS-Clip, finding that most of its target sequences were intronic. Using previously available HITS-Clip data from HSA^LR^ mice [38], an overlap of 188 common transcripts was found between MBNL1 (1351 transcripts), MBNL2 (523 transcripts), and MBNL3 (842 transcripts) targets, suggesting that MBNL proteins have both common and unique binding sites. For example, the *Calm3* transcript presents overlapping binding sites for all three MBNL proteins in two different polyadenylation sites. To clarify the biological effects of APA in DM1, HSA^LR^ quadriceps muscle was analyzed using polyA-Seq. The findings suggest that MBNL sequestration by CUG repeats often leads to reversion to the fetal APA pattern observed in MEFs. To validate these observations, human DM1 and control vastus lateralis were analyzed by both polyA-Seq and microarray. Similar to findings in DKO/3KD MEFs, the results showed that DM1 patients have a majority (59%) of polyadenylation sites shifted upstream. Key genes related to muscle atrophy (e.g., *IGF-1*, *HDAC5*, and *mTOR*) showed altered APA, suggesting that pathways known to inhibit protein synthesis and activate protein catabolism were altered [9].

Proteomics studies provide critical data about protein post-translational modifications, activity, and stability, which are crucial to elucidate the cellular pathways affected in DM1. Since these effects cannot be reliably predicted from RNA-Seq data, a combination of both transcriptomics and proteomics is desirable. In this regard, André et al. used myoblasts derived from DM1 patients (carrying a (CTG)2600 repeat expansion) with isogenic controls obtained after removal of the repeat tract by CRISPR/Cas9 gene editing [49]. Fifty-three proteins were differentially expressed when comparing myoblasts with and without the repeats, including two members of the metallothionein family of proteins, MT1L and MT2A, which were downregulated together with several other myogenesis-related proteins. MBNL1 protein levels were also strongly downregulated despite remaining unchanged at the mRNA level, something that the authors validated in a later study [49].

### 2.2. Omics Studies in the Central Nervous System

Central nervous system (CNS) impairments are also prominent in DM1, particularly in congenital cases, which present with intellectual disability, attention-deficit hyperactivity, and autism spectrum disorders [50]. The broad and heterogeneous cognitive and neuropsychological profiles in DM1 suggest the involvement of multiple brain areas and neuronal circuits, with defective neuron–glia interactions. Several aspects fundamental for proper brain function are likely affected and contribute to DM1 pathophysiology [26,51].

An important question addressed in the past was whether MBNL proteins are functionally affected by expanded CUG-repeats in the diseased brain, as suggested by the colocalization of *DMPK* RNA foci with MBNL1 and MBNL2 in brain samples [52,53]. Indeed, an RNA-Seq approach using frontal cortex and hippocampus samples from DM1, DM2, and control autopsies revealed 596 alternative exons with lower inclusion and 335 exons with higher inclusion corresponding to MBNL2 direct targets identified by HITS-Clip. Furthermore, polyA-Seq and RNA-Seq experiments were performed, identifying 2826 genes with MBNL2-regulated APA events, of which 56 were also regulated by MBNL2 at the AS level. Additionally, most APA alterations were similar to the ones described in the skeletal muscle. Altogether, the use of these three approaches confirms that MBNL2 becomes depleted from its normal RNA targets, having downstream effects on AS and APA [54].

In a complementary approach, a glial cell model of DM1 was generated using the MIO-M1 (human Muller glia) cell line expressing human *DMPK* gene constructs with 0 and 648 CTG repeats in an inducible doxycycline-dependent manner. Control and DM1 cell lines were analyzed using microarray technology to quantify gene expression, finding alterations in the inflammatory response. Furthermore, MBNL1- and MBNL2-dependent AS were dysregulated, and both splicing factors colocalized with CUG RNA foci, demonstrating that in MIO-M1 cells, lacking the natural *DMPK* genomic context, CTG expansions are sufficient to recapitulate critical molecular aspects of the disease [55].

A few recent studies have combined multiple omics approaches, including proteomics analysis, to elucidate DM1 pathological mechanisms in the brain using the DMSXL mouse model, with characteristic disease alterations in the CNS [24,25]. In a first study analyzing the frontal cortex and hippocampus of DMSXL mice, SYN1 proteins were found hyperphosphorylated in neurons by comparing the proteomic profile of the DM1 model with those of DM20 control mice expressing normal-sized *DMPK* transcripts [26]. Using DM20 controls for comparison allowed the proteomics data to cancel out disease intermediates originating from *DMPK* overexpression rather than expanded CUG RNA. SYN1 aberrant activation by phosphorylation affects synaptic vesicle transport, which was confirmed in transfected cells and post-mortem brains of DM1 patients. In another study analyzing cerebellum samples of the same mouse model, a global proteomics approach identified strongly downregulated GLT1 (a glutamate transporter) in DM1 mice and human patients [51]. GLT1 downregulation in DM1 astrocytes is caused by MBNL depletion and results in increased glutamate neurotoxicity, neuronal death, and motor incoordination. In a more recent study using the DMSXL mouse model, the authors used a multi-omics approach to investigate alterations occurring during the cell culture differentiation of four brain cell types (neurons, astrocytes, oligodendrocytes, and oligodendrocyte precursor cells), showing that astrocytes present the largest number of splicing alterations [56]. Moreover, 85 genes were observed with significantly different expressions in oligodendrocytes, of which the vast majority were downregulated. Next, they analyzed the shift in gene expression during the differentiation of oligodendrocytes compared to WT cell cultures. Interestingly, 80 of the 85 differentially expressed genes were altered during oligodendroglia maturation, suggesting that expanded CUG RNA primarily affects the expression of genes that are regulated during the differentiation of this cell type. In addition, the transition from an embryonic to a mature splicing pattern was also delayed in both oligodendrocytes and astrocytes, compared to the corresponding WT cells. These results suggest the presence of impaired programs of DM1 neuroglia differentiation in vitro. Functional annotation of the affected genes yielded Gene Ontology (GO) terms related to cell-membrane-dependent processes, such as cell adhesion, morphogenesis, and extracellular matrix. An exon ontology analysis of all exons abnormally spliced in astrocytes revealed enrichment for functions related to protein phosphorylation and localization, which was validated experimentally with some candidates.

A study in the frontal cortex comparing the transcriptome from 21 DM1 and 8 unaffected post-mortem, age- and sex-matched individuals found 130 exons differentially included between DM1 and controls [57]. These alterations were related to key synaptic scaffolding proteins, cytoskeletal organization components, ion channels, and neurotransmitter receptors. Analyses of cis-elements flanking dysregulated exons revealed that MBNL-bound motifs were the predominant signature, indicating that functional depletion of MBNL may be a major driver of splicing changes in the frontal cortex of DM1 patients. Results were compared to previous reports from muscle biopsies of *tibialis anterior* and heart muscles [42], finding a solid correlation between 25 commonly dysregulated exons, including *MBNL1* exon 5, *MBNL2* exon 5, and *MAPT* exon 3, suggesting a similar mechanism for splicing dysregulation across all tissues [57].

### 2.3. Omics Studies in the Heart

Another highly studied tissue in DM1 is the heart. Alterations specific to this organ were described in a transcriptome analysis using AS profiling microarrays performed in embryonic and adult MBNL1^ΔE3/ΔE3^ mice [10]. This work identified no correlation between AS and gene expression in either developmental stage but found enriched binding motifs in flanking introns of developmentally regulated exons for CELF, MBNL, and Fox RBPs. The study also measured protein levels and gene expression during embryonic development and the first two weeks after birth, observing a decrease in CELF levels and an increase in MBNL1 protein levels. Interestingly, the data showed that Fox-1 levels were negatively correlated with the depletion of CELF1 protein, while CELF2 protein decreased when Fox2 levels dropped and MBNL levels began to rise, implying a complex relationship between these RBPs. Consistent with the developmental modulation of MBNL and CELF at the protein level, 10 AS transitions correlated with this developmental shift. Collectively, these analyses identify and characterize a highly regulated AS program that supports cardiac tissue postnatal growth [10].

A similar study was performed in the hearts of *Gallus gallus* to identify transcripts regulated by CELF1 in the embryonic myocardium. RNA-Seq and CLIP-Seq analyses were performed for this purpose, finding that 75% of the CELF1 CLIP tags within genes mapped to intronic regions, consistent with the localization of CELF1 in the nucleus in embryonic heart muscle cells, and with its known role as a regulator of pre-mRNA AS in the heart [58]. Upon overexpressing CELF1 in mouse hearts, the authors found 234 splicing events that were altered by this RBP when compared with the previous splicing microarray data available, supporting their previous observations [38]. Of the 120 events regulated by CELF1 and MBNL1, 78 were regulated in an antagonistic fashion, suggesting that the regulated inclusion of exons tended to oppose the normal developmental transitions upon MBNL1 depletion and CELF1 induction. Interestingly, CELF1 induction in the heart was produced by only modest changes (around 15% higher) in mRNA levels [59].

Failure of a developmental switch was also found in the heart-specific EpA960 mouse model that inducibly expresses human *DMPK* exon 15 containing 960 CUG repeats. A coordinated adult to embryonic shift was observed in the hearts of these mice upon CUG repeat RNA expression, as analyzed by microarray technology. Functional annotation of the altered genes revealed downregulation in the cardiac transcription factors *MEF2a* and *MEF2c*, correlating with the mRNA and miRNA alterations described in this tissue [60]. In the same study, MEF2 was confirmed to be reduced in human cardiac samples. Exogenous MEF2C rescued MEF2 target miRNA and mRNA normal expression patterns in a cell model of cardiac disease. To further evaluate the role of CELF1, RNA-Seq was used to analyze DM1 heart, human fetal heart, and non-affected adult heart samples, demonstrating that RNA-Binding FOX2 (*RBFOX2)* was up-regulated in this DM1 tissue. Next, a doxycycline-dependent CELF1 overexpression mouse was developed to understand the relationship between this RBP and CELF1. This model showed a significant change in *RBFOX2* isoforms and pro-arrhythmic phenotypes, demonstrating a link between elevated CELF1 activity and DM1-related cardiac conduction delay and arrhythmogenesis [61].

To evaluate the extent of the gene expression alterations present in DM1 hearts, an RNA-Seq approach compared AS events in heart and *tibialis* tissues from the autopsies of 10 DM1 patients with three unaffected individuals [42]. The findings of this study showed that exons of the cytoskeleton-related gene *PDLIM3* were altered in all DM1 samples, suggesting a possible biomarker for disease progression in this tissue. A high correlation was found in the percentage of cassette exon inclusion between the heart and *tibialis*, suggesting that the molecular mechanisms of DM pathogenesis are conserved across both tissues. Many of these exons have been validated as MBNL-dependent targets, including *MAPT*, *NUMA1*, *MBNL2*, *NCOR2*, and *LDB3*. Nevertheless, several exons were tissue-specific for heart or *tibialis* [42].

Recently, a new mouse model has been reported, with inducible expression of human *DMPK* exons 11–15 carrying 960 interrupted CTG repeats in cardiomyocytes [62]. RNA-Seq from heart samples in this model revealed gene expression and AS changes in ion transport genes associated with inherited cardiac conduction diseases, including a subset of genes involved in calcium handling. Consistent with the RNA-Seq results, calcium-handling defects were identified in atrial cardiomyocytes isolated from this model, potentially contributing to the observed arrhythmogenic phenotypes [62].

### 2.4. Omics Studies in Other Affected Tissues in DM1: Lens and Thymus

Misregulated AS and alterations in RBP expression explain key phenotypes in the muscles, brain, and heart, but DM1 clinical manifestations in other tissues remain poorly understood. One example where omics approaches are providing insight is in explaining cataracts. Recent studies using microarray technology with epithelial cell lines established from the cataracts of affected DM1 and DM2 patients suggest that *DMPK* mRNA is highly expressed in the lens epithelia, which indicates that significant RNA toxicity may be occurring in the eyes. Gene expression alterations were also highly similar in DM1 and DM2 samples, identifying 317 significantly altered common genes and no significant alterations in *DMPK* transcript levels between control and DM1 or DM2 samples. A pathway analysis of the altered common genes in DM1 and DM2 identified type 1 interferon signaling pathways as a potential contributor to the advent of cataracts [63,64].

Although traditionally neglected in the DM1 field, the immune system is also known to be affected in DM1 [65]. In one study, mouse thymus development and its role in DM1 were analyzed [66]. Using RNA-Seq, the authors found *MBNL1* to be highly expressed while *DMPK* had little expression during embryogenesis, suggesting that MBNL proteins play a vital role in thymic organogenesis and thymocyte development. To further evaluate this hypothesis, publicly available RNA-Seq data from 9-week-old MBNL1^ΔE3/ΔE3^ mice were compared to littermate controls, finding over 1000 genes with altered expression in pathways related to the apoptotic process and cell differentiation. At the AS level, 866 alterations were found, including known MBNL1 targets *CLASP1*, *NCOR2* exon 46, *DNM2* exon 10 and 11, and transcription factor *Lef1* exon 6, which has been extensively described as an important factor in thymocyte development [67,68]. They also used MBNL1 siRNA knockdown in C2C12 mouse myoblasts, which was compared to control cells by CLIP-Seq to evaluate whether the splicing defects seen in *Lef1* are related to MBNL1 depletion. Results indicate that MBNL1 proteins play a role in pre-mRNA processing by regulating the AS of these critical transcription factors required for normal T-cell development [66].

### 2.5. Other Omics Studies in DM1

Given their previously known significance in human disease, DNA methylation (DNAme) and regulatory microRNA (miRNA) profiles are suitable for characterization in DM1 by means of whole-genome epigenomics experimental strategies. To date, DNAme omics studies in DM1 have focused only on CpG islands surrounding the CTG repeat [69], mainly approached using pyrosequencing omics methodology.Accumulating evidence supports that hypermethylation levels are associated with DM1 pathogenesis, with different tissues and clinical forms being affected [69]. Only one genome-wide methylation study has been performed in DM1, involving also several other trinucleotide repeat disorders. DM1 patient samples did not show a significant increase in global 5-mC compared to age- and sex-matched unaffected individuals, demonstrating the validity of targeting only specific CpG regions in DM1 [70]. Recently CpG methylation has been studied in differentiation stages, comparing hESC, myoblasts, and myotubes from control and DM1 samples. Findings suggest that congenital DM1 presents CpG methylation in the upstream site and that they increase as differentiation continues [41].

MiRNAs are ncRNA molecules that play crucial roles as epigenetic modulators, inducing post-transcriptional changes in gene expression by mRNA destabilization and/or translation repression [71]. The expression of several miRNAs has already been reported as altered in DM1, most quantified by RT-qPCR in mice or human cells (recently reviewed in [72]). Specifically based on omics experimental approaches, an RNA-Seq study was performed in patient-derived cells that analyzed the miRNA/mRNA interactions as related to the enrichment of RNA-induced silencing complex (RISC)-bound miRNAs. The study identified 24 correlations, of which 22 were first described in this work [73]. As an example of the validity of transcriptomics as a biomarker discovery tool, several miRNAs have recently been identified as DM1 biomarkers in patient hearts, skeletal muscle, and serum using RNA-Seq. A subset of miRNAs (miR-223-3p and miR-24-3p) was also found to be increased in DMSXL mice serum. Furthermore, miR-223-3p, which has been described as a neural cell degradation protector, was significantly reduced in the brain of DMSXL mice [74].

DM1 is also defined as a metabolic syndrome due to features such as dyslipidemia, insulin resistance, hypertension, and pro-inflammatory state, among other alterations consistently present in DM1 patients [75]. Recently, technologies such as proton nuclear magnetic resonance (NMR) and mass spectrometry (MS) have enabled major metabolomics studies to identify significant changes in specific metabolites in DM1. In one study, while branched-chain amino acids, acetate, and creatine levels were found elevated in the serum of patients with DM1, lysine and guanidine acetic acid levels were reduced, similar to observations in other muscular diseases [76,77]. Distinctively, glutamine levels were elevated only in the serum of DM1 patients, unlike the other muscular pathologies studied (where they were reduced), pinpointing this metabolite as a promising biomarker in clinical neurologic outcomes in conditions such as DM1 [72].

## 3. Omics Studies in Candidate Drug Validations

Given the well-defined transcriptome alterations associated with DM1, evaluating the degree of rescue and off-target effects of candidate drugs during preclinical testing by AS quantification has become customary in the field. Among the therapeutic strategies currently being explored [12,78], the repurposing of already-known small molecule drugs and RNA-based approaches has gained increasing attention for genome-wide RNA-Seq approaches, as discussed below.

In a recent study, autophagy was found to be up-regulated in DM1 patient muscle biopsies [79] and PDIMs [80], which led to the hypothesis that repressing this catabolic pathway could improve muscle wasting. Chloroquine is a well-known autophagy blocker [81], and treatment of PDIMs with this drug improved transcriptome level alterations as measured by RNA-Seq, leading to 59% recovery in disease-related genes in GO terms related to muscle homeostasis and function. Crucially, results were validated in the HSA^LR^ mouse model without signs of toxicity. One of the most relevant rescue mechanisms was the increase of Muscleblind protein levels in *Drosophila*, HSA^LR^ mice, and PDIM disease models, which prevented protein degradation [82].

An alternative approach, used by Nakamori et al., was to interrupt the interaction between toxic RNA and MBNL proteins. A drug screening searching for small molecules with these characteristics found erythromycin as a promising candidate [83]. Using HSA^LR^ mice versus controls, a synergistic effect was found when erythromycin was combined with furamidine. RNA-Seq analysis detected recovery of around 73% of the AS defects observed in HSA^LR^ mice, a percentage nearly double the amount of that detected for furamidine or erythromycin alone, independently of dosage. Off-target effects of each treatment were evaluated by quantifying the alterations not present in control vs. HSA^LR^ mice but present in the HSA^LR^ vs. HSA^LR^-treated comparison. Off-targets were increased in the combined treatment compared to each treatment alone, but not nearly as much as when higher doses of each drug were tested alone [84].

A type 2 diabetes mellitus drug known as metformin is well ahead in the race to develop a treatment for DM1. It was tested in mesodermal precursor cells differentiated from control and patient embryonic stem cell lines at 10 and 25 mmol/L, following which 63 and 1171 genes showed recovery of expression levels, respectively. Regarding AS correction, 95 and 416 exons presented a reversal above 10% at each dose. Gene enrichment analysis of the 416 transcripts identified gene sets involved in the cytoskeleton, nuclear lumen, and RNA binding. A subset of recovered AS events were validated by RT-qPCR, including DM1-related mis-splicing of *INSR* exon 11, *TNNT2* exon 5, *ATP2A1* exon 22, *DMD* exon 71, *DMD* exon 78, and *KIF13A* exon 32, among others [85].

Since DM1 is a genetic disease, RNA-based therapies are potentially an effective strategy. An approach tested in one study aimed to degrade toxic *DMPK* transcripts via activation of the RNase-H machinery in the nucleus using antisense oligonucleotides (ASOs). Microarray technology was employed to evaluate alterations in genes related to apoptosis, cytotoxicity, and inflammation processes at tenfold the optimal dosage, finding no significant alterations in these processes. Promising results were also found rescuing the expression of 41 and 14 genes affected in human cell culture and DMSXL mice, respectively, advancing progress towards a clinical trial [86].

miR-218 has been reported as a natural repressor of MBNL1 and 2 [87], and it has been found overexpressed in patient-derived samples and cell models [88]. Recently, chemically engineered antisense oligonucleotides against this miRNA, also called antagomiRs, have been described as a candidate therapy for DM1 because of their potential to upregulate endogenous levels in MBNL1 and 2, thus compensating for sequestration by expanded CUG repeat RNA. Using an RNA-Seq approach with in vitro differentiated DM1 myotubes, miR-218 inhibition was found to rescue up to 34% of overall gene expression alterations. Although the analysis did not include AS, several critical genes improved their expression upon antagomiR treatment. Among others, MBNL1 and 2 levels significantly increased compared to control antagomiR-treated cells, while MBNL3, previously involved in myogenesis inhibition [15], was repressed. Furthermore, the levels of *GSK3B* (known to contribute to muscle atrophy [89]) and *DMPK* were significantly downregulated upon treatment, suggesting a positive impact on known therapeutic targets of the disease.

CRISPR/Cas9 is a promising platform to potentially correct dominant genetic diseases by gene editing, showing unprecedented precision. In the DM1 field, this technology was applied to patient-derived iPSC to delete the expanded CTG repeats, after which these cells were compared to non-corrected ones by RNA-Seq [90,91]. The authors found less than 1% (fewer than 200 genes) with significant differences in gene expression (reads aligned to genes), while there were over a thousand significant differences in transcript expression (reads aligned to gene isoforms). The 99 most significant altered transcripts were selected and functionally annotated, revealing altered pathways related to cardiac development, maturation, and function, including the expression of gated ion channels. Interestingly, 7 out of 99 top-affected transcripts were involved in AS, as splicing regulators or spliceosomal complex components. AS was also investigated, revealing a difference in the splicing pattern of genes associated with cardiac function, cellular signaling, and DM1 disease markers such as *NUMA1*, *MBNL2*, and *LDB3*. Interestingly, disease-causing *DMPK* transcripts also presented AS.

## 4. Conclusions and Future Perspectives

This review has provided an overview of the most relevant key studies to date using omics data to elucidate DM1 molecular pathogenesis (Table 1), putting into perspective the specific alterations discovered in each tissue, the methodologies used to identify them, and the contribution of each study to the development of effective DM1 treatment.

As known major players in DM1 pathology, MBNL and CELF proteins are a primary focus of analysis in multiple omics studies. Mounting omics evidence suggests critical roles for these RBPs as competitors or co-regulating factors in key AS and APA events. The ubiquitous nature of the *DMPK* gene, together with the toxicity of expanded CUG repeats over both RBPs, may partly explain the multisystemic nature of DM1. In addition, the key role in splicing regulation of these proteins most probably accounts for the high quantity of AS alterations found across DM1 tissues.

Future perspectives in DM1 research advocate implementing current state-of-the-art technologies such as single-cell and third-generation sequencing. Single-cell analysis is currently used as a potent tool to explore omics changes in tissues composed of a complex mixture of cell types. This could be particularly important in the DM1 field since the most affected tissues are composed of a variety of cells with very specialized functions. Furthermore, expanded CTG microsatellites show somatic instability [92], presenting genetic mosaicism in each tissue. The cells usually carry a different number of CTG repeats in their genome, thus contributing to an intrinsic degree of variability. Therefore, the findings of most of the studies described herein can only be discussed in the context of pooled information coming from these mixtures of cells, which in many cases might obscure important findings and limit the scope of the study. Studying omics from cells differentiated in vitro can partially overcome this problem but does not provide the same context and clues as to the conditions present in vivo, constituting a fundamental limitation of this approach. To this end, single-cell analysis of DM1 tissue could provide important new information to elucidate how the disease affects the function of each cell type in the original context.

The unprecedented accuracy in evaluating the RNA composition of DM1 samples has provided strong support for the toxic RNA hypothesis, by pinpointing all AS and APA changes in the disease and the regulatory factors involved. Moreover, omics approaches now provide global metrics to compare the efficacy of candidate therapies. Nevertheless, current sequencing technologies suffer from limitations: repeat expansions or structural variants have proven very challenging to sequence, and, despite certain progress, there is still room for improvement. Third-generation sequencing, such as Oxford Nanopore Technologies (ONT) or Pacific Bioscience single-molecule, real-time technologies (SMRT), can bypass these limitations by generating long reads covering an average of 20 Kbp and 30 Kbp, respectively, with the current record being 2.3 Mbp of a single continuous read from ONT [93,94,95,96]. For DM1, this could open up interesting new areas of research for an accurate description of RNA isoform composition, instead of exon use as the current state-of-the-art, with major implications for diagnostic, structure, and drug discovery applications. As was seen recently in the evaluation of CpG islands methylation [41].

Another field with great potential is data mining of already available datasets. For example, taking advantage of the extreme depths of RNA-Seq data in dmseq.org [42], one study analyzed the misexpression of circular RNAs (circRNAs: highly stable, lowly expressed RNAs coming from protein-coding transcripts). Although their origins are still under-researched, several RBPs are known to participate in their biogenesis [97]. The average expression of specific circRNAs, such as circCAMSAP1, circHIPK3, circNFATC3, and circZKSCAN1, is increased in DM1 with no similar change in the linear transcripts from which they derive. Interestingly, a negative correlation between circRNA level in *tibialis anterior* tissue and ankle dorsiflexion force was found, suggesting a relationship between circRNA and disease severity [98].

Finally, breakthrough advances in artificial intelligence (AI) are enabling accurate protein structure predictions [99], which in the context of DM1 research could help anticipate the effects of specific AS alterations at the proteome level and integrate multi-omics approaches into our understanding of the disease. Taken together, this technology could support researchers in their goal to enhance knowledge of the intrinsic behavior of DM1-affected proteins, through which new types of structure-specific treatments can be developed to correct molecular damage.

## Figures and Tables

**Figure 1 ijms-23-01441-f001:**
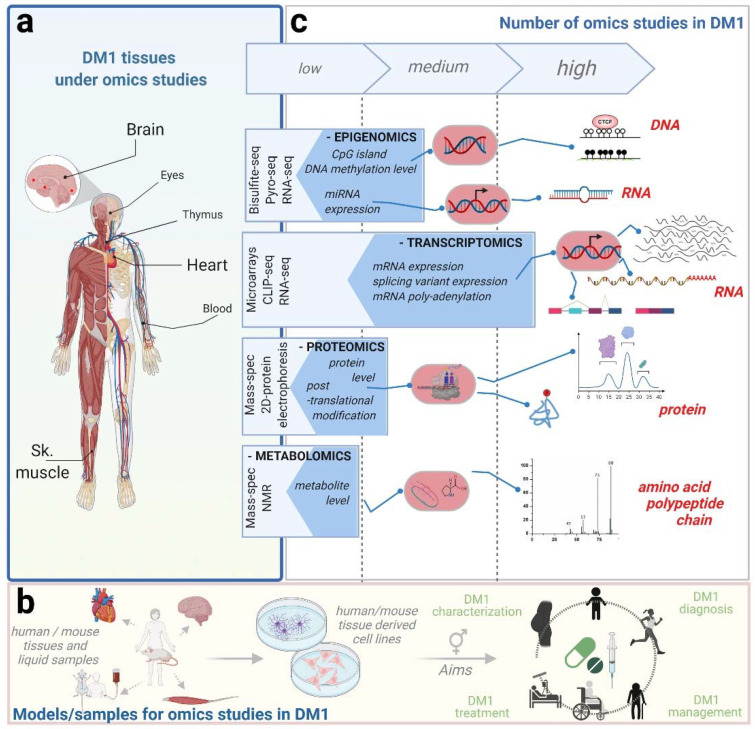
Omics approaches in DM1 research. Research in DM1 relies increasingly on omics-based technologies to decipher molecular pathogeneses in key affected tissues, such as skeletal muscle, brain, and heart (**a**). Experimental conditions aim to elucidate certain aspects of DM1 pathology and include candidate drug testing on human primary cells, isolated directly from DM1 patient tissues: these retain the morphological and functional characteristics of their tissue of origin but display limited potential for self-renewal and differentiation. Solid omics approaches have therefore also been performed using human immortalized cell lines, as well as samples from different mouse DM1 models (**b**). Most frequent approaches focus on the transcriptome (RNA-Seq, CLIP-Seq, and microarrays) to build a comprehensive view of DM1 disease status at different levels of RNA metabolism (**c**). Furthermore, an increasing number of studies using epigenetics, proteomics, and metabolomics approaches are providing novel insights into altered pathways in DM1 (**c**). Altogether, these technologies provide valuable data for fundamental studies in DM1 but also on translational research characterizing drug mechanisms of action. Created with BioRender.com (accessed date 20 December 2021).

**Table 1 ijms-23-01441-t001:** Current omics data available from DM1 studies.

Technique	Type	Sample	Species	Objective	Reference	Cite
CLIP-Seq	Illumina Genome Analyzer II	Embryonic fibroblasts	Human, mouse	Alternative polyadenylation	GSE60487	[9]
2D WB and nano LC-MS/MS	MALDI-TOF Voyager -DE-STR and Q-TOF MS	Frontal cortex and hippocampus	Mouse	Proteome and phosphoproteome analysis	Not provided	[26]
Microarray	ares_ucsc_mouse_59198_affyMouseA	Skeletal muscle biopsies	Mouse	Alternative splicing	GSE17986	[36]
RNA-Seq	Illumina HiSeq 2000	Brain, heart, skeletal muscle, and cell cultures	Mouse	Transcriptome analysis	GSE39911	[38]
CLIP-Seq	Illumina Genome Analyzer II	Brain, heart, skeletal muscle, and cell cultures	Mouse	RBP binding sites	GSE39911	[38]
RNA-Seq	Illumina HiSeq 2500	Quadriceps	Mouse	Transcriptome analysis	PRJNA625451	[40]
RNA-Seq	Illumina NextSeq 500	Embryonic stem cells, myoblast, myotubes	Human	Transcriptome analysis	GSE160916	[41]
RNA-Seq	Illumina HiSeq 2000	Skeletal muscle and heart biopsies and autopsies	Human	Transcriptome analysis	GSE86356	[42]
Microarray	Human Exon 1.0 ST array	Skeletal muscle biopsies	Human	Alternative splicing	GSE48828	[47]
LC-MS/MS	EASY-nLC 1200-rbitrap Tribid MS	Myoblasts	Human	Global proteome analysis	PXD016056	[49]
iTraq Nano-LC-MS/MS	Ultimate 3000 RSLC LTQ-Orbitrap Velos MS	Cerebellum	Mouse	Global proteome analysis	Not provided	[51]
CLIP-Seq	Illumina Genome Analyzer II	Brain	Human, mouse	RBP binding sites, alternative polyadenylation	GSE68890	[54]
Microarray	Clariom D Arrays	Cell culture	Human	Transcriptome analysis	GSE164057	[55]
RNA-Seq	Illumina HiSeq 2500	Astrocytes, oligodendrocytes, and neurons	Mouse	Transcriptome analysis	GSE162093	[56]
2D WB and nano LC-MS/MS	Nano RSLC-Q	Astrocytes	Mouse	Phosphoproteome analysis	PXD025011	[56]
RNA-Seq	Illumina NextSeq 500	Frontal cortex biopsies	Human	Transcriptome analysis	GSE157428	[57]
CLIP-Seq	Illumina Genome Analyzer II	Heart	Chicken	RBP binding sites	GSE67360	[58]
RNA-Seq	Illumina HiSeq 2000	Muscle, heart	Mouse	Transcriptome analysis	GSE61893	[59]
CLIP-Seq	Illumina Genome Analyzer II	Muscle, heart	Mouse	RBP binding sites	GSE61893	[59]
Microarray	Illumina MouseWG-6 v2.0 expression beadchip	Heart	Mouse	Transcriptome analysis	GSE48991	[60]
RNA-Seq	Illumina HiSeq 4000	Heart	Mouse	Transcriptome analysis	GSE126771	[61]
RNA-Seq	Illumina NovaSeq 6000	Heart	Mouse	Transcriptome analysis	GSE164825	[62]
Microarray	HumanHT-12 v3 Expression BeadChip	Lens epithelial	Human	Transcriptome analysis	E-MEXP-3365	[64]
RNA-Seq	Illumina HiSeq 2500, Illumina NextSeq 500	Thymus	Human, mouse	Transcriptome analysis	GSE138691	[66]
RNA-Seq	Illumina HiSeq 2500	Biceps brachii	Human	miRNA/mRNA interactions	GSE108592	[73]
RNA-Seq	Illumina NextSeq 500	Blood Muscles, heart, and brain	Human, mouse	miRNA analysis	PRJEB46413	[74]
RNA-Seq	Illumina NextSeq 550	Cell culture	Human	Transcriptome analysis	GSE128844	[82]
RNA-Seq	Illumina NextSeq 500	Quadriceps	Mouse	Transcriptome analysis	PRJNA555349	[84]
RNA-Seq	Illumina HiSeq 2000	Cell culture	Human	Transcriptome analysis	GSE138789	[85]

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
