# Peer review of "Deciphering the Complex Molecular Pathogenesis of Myotonic Dystrophy Type 1 through Omics Studies"

_ijms, 2022, doi:10.3390/ijms23031441_

Round 1

Reviewer 1 Report

This is a comprehensive, updated and well-structured review on the omics studies in myotonic dystrophy type 1. 

The text, figures, tables and references are adequate. 

I found it an interesting read, relevant to the field and with important considerations for future application. 

A minor comment regards the title "deciphering the complex molecular pathogenesis of myotonic dystrophy through omics study". I expected also studies on myotonic dystrophy type 2 which has not been covered in this paper. I would suggest to change the title, specifying "myotonic dystrophy type 1" and explaining than in the introduction why DM2 has not been covered. 

Author Response

Dear Reviewer 1, thank you for your time and effort in reviewing this manuscript. We appreciate your comments and thoughtful insights.

We have decided to alter the title to “Deciphering the complex molecular pathogenesis of myotonic dystrophy type 1 through omics studies”, and add “Herein, we reviewed the literature on DM1 since the number of omics studies far exceeds those involving Myotonic Dystrophy Type 2 (DM2), focusing particularly on the tools used to elucidate the pathological mechanisms involved, to broaden the scope of our understanding of this complex multisystem disorder.” On lines 90 to 93 to justify the exemptions of DM2 only omics studies as you have kindly suggested.

Reviewer 2 Report

The manuscript „Deciphering the complex molecular pathogenesis of myotonic dystrophy through omics studies” provides an interesting insight into the molecular pathogenesis of myotonic  dystrophy. In my opinion, the presented review describes in an interesting way contemporary achievements in the field of possibility of studying the complexity of the pathogenesis of the incurable disease, such as myotonic dystrophy, which may facilitate the treatment of this disease in the future. The article also emphasizes the importance of the information described and the possibility of using them in future perspectives for the development of knowledge about the disease and its treatment. 

Minor comments:

In the article, it is possible to emphasize the importance of the disease, especially its incurability, and mainly the fact that it is treated only symptomatically. Therefore, it is worth adding a few words about the current methods of treatment and medical diagnosis of this disease. That would enrich the work additionally.

Moreover, the description under Figure 1 lacks the word "Figure 1":

FIGURE 1. "Omics approaches in DM1 research. Research in DM1 increasingly relies on omics-based 108 technologies to decipher molecular pathogenesis in key affected tissues, such as skeletal muscle, 109 brain, and heart (a). The experimental conditions aim to explain or tackle certain aspects of DM1 110 pathology and the testing of candidate drugs on human primary cells, isolated directly from DM1 111 patients´ tissues -retaining the morphological and functional characteristics of their tissue of origin, 112 but displaying limited potential for self-renewal and differentiation-. Thus, solid omics approaches 113 have also been performed using human immortalized cell lines, as well as under usage of samples 114 from distinct mouse DM1 models (b). Most frequent approaches focus on the transcriptome (RNA- 115 seq, CLIP-seq, and microarrays) to build a comprehensive view of DM1 disease status at different 116 levels of RNA metabolism (c). Furthermore, an increasing number of studies using epigenetics, 117 proteomics, and metabolomics approaches are providing novel insights into altered pathways in 118 DM1 (c). Together, these technologies are providing valuable data to fundamental studies in DM1, 119 but also on translational procedures regarding the characterization of a drug´s mechanism of action".

In my opinion, the presented is an interesting summary of the current state of knowledge for future researchers and I recommend the manuscript for publication after making appropriate corrections and changes.

Author Response

Dear Reviewer 2, thank you for your time and effort in reviewing this manuscript. We appreciate your comments, thoughtful insights, and recommendations, they have been addressed in lines 42 to 48, hoping to cover your suggestion and the text correction has been added to Figure 1.

Reviewer 3 Report

Summary: This review focuses on the omics studies in the field of myotonic dystrophy. The authors provide a summary on the omics studies in skeletal muscle, heart and in the central nervous system. The authors also summarize other omics studies in DM1. Overall, this is a very comprehensive review of the literature on omics in myotonic dystrophy. It is well written and will be a very good resource for those interested in learning about the omics studies so far done in DM1 field. Minor comments: In the Figure showing omics approaches in DM1 research, single-cell sequencing approach should also be mentioned and their advantage. Although, no one has used this technology in DM1 yet. This approach also gives on cellular composition of diseased tissues. On page 8, please check the citation for reference #63. Page 10 of the review, last paragraph, In the DM1 field, this-----, this sentence repeated twice.

Author Response

Dear Reviewer 3, thank you for your time and effort in reviewing this manuscript. We appreciate your comments and recommendations.

We are grateful for your sharp insight in that reference 63 was misplaced, which was corrected in both the main text body and references. Page 10 correction has also been addressed.

As you have noted, single-cell sequencing has not been used in DM1 yet. Figure 1 tries to show all currently used techniques in DM1, and that is why we have decided to add it to the future perspectives on page 12, part 4, second paragraph, where we cover the possible benefits of using this technique in the future.